# Structured Evaluation of Synthetic Tabular Data

## Abstract

Tabular data is common yet typically incomplete, small in volume, and access-restricted due to privacy concerns. Synthetic data generation offers potential solutions. Many metrics exist for evaluating the quality of synthetic tabular data; however, we lack an objective, coherent interpretation of the many metrics. To address this issue, we propose an evaluation framework with a single, mathematical objective that posits that the synthetic data should be drawn from the same distribution as the observed data. Through various structural decomposition of the objective, this framework allows us to reason for the first time the completeness of any set of metrics, as well as unifies existing metrics, including those that stem from fidelity considerations, downstream application, and model-based approaches. Moreover, the framework motivates model-free baselines and a new spectrum of metrics. We evaluate structurally informed synthesizers and synthesizers powered by deep learning. Using our structured framework, we show that synthetic data generators that explicitly represent tabular structure outperform other methods, especially on smaller datasets.

## 1 Introduction

Tabular data is among the most common and versatile data formats for data science and machine learning. Compared to text and imagery, tabular data are often incomplete, imbalanced, and small in volume because they can be expensive and difficult to collect. Even upon collection, the data may be limited in access due to privacy concerns. Synthetic data generation can alleviate or even solve the above problems. A good data synthesizer is a generative model that learns the actual data generating process or distribution. As such, the data synthesizer can be queried to produce as much data as desired. Furthermore, a desirable data synthesizer can predict the values (including missing values) of any set of columns conditioned on any non-overlapping set. This ability allows the synthesizer to generate balanced datasets, impute missing values, and debias the data.

To know whether a synthesizer is effective, it is crucial to have a coherent and complete set of evaluation metrics. Figure 1A shows a modern taxonomy of the evaluation metrics. On the first level, the metrics are arranged into model-based versus model-free. Model-based metrics, also called "likelihood tests," evaluate the synthetic data by computing its likelihood under the known, ground-truth data generating process (e.g. in Xu et al., 2019). The model-free metrics are further divided into resemblance fidelity and application fidelity (Dankar et al., 2022). The latter, often referred to as "machine learning (ML) efficacy," is the most popular type of metric. While this taxonomy provides a methodological organization of the metrics, we still do not know how they are related (coherence) and whether we are missing important aspects of the evaluation (completeness). To address this issue, we propose a framework that interprets the metrics under a unifying objective and repositions them along a spectrum of structure (Figure 1B).

Our framework stems from a formal objective that a synthesizer should produce *samples from the same joint distribution* as the real data. We show how each metric can be derived from this core objective by identifying the substructure of the joint distribution targeted by the metric. This analysis reveals the relationships among the metrics and their completeness: that is, each metric targets a particular aspect of the joint distribution, and the targeted aspects span the full range of distributions from simple marginals to the full joint (Figure 1B). Without the objective and the structure, it is unclear what even constitues a complete set of evaluation metrics. The structured framework also

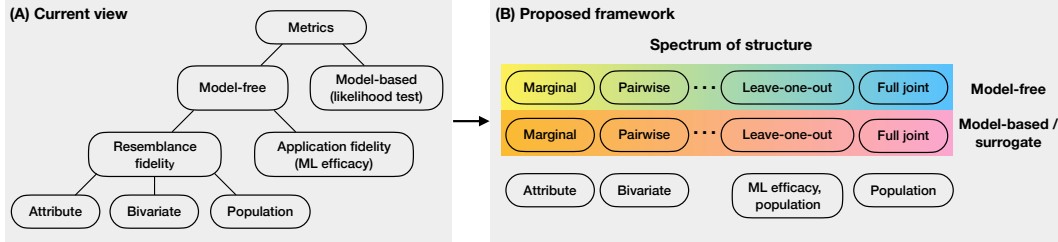

Figure 1: (A) A modern taxonomy of the evaluation metrics. Some of the naming conventions follow Dankar et al. (2022). (B) A structured framework of evaluation metrics. The metrics are positions along a spectrum of structure, depending on the structure of distribution they target. This spectrum is applied to both model-free and model-based metrics. The bottom row shows where the metrics depicted in (A) are repositioned in the new framework.

motivates model-free baselines that are easy-to-interpret (Section 3.2). We further introduce a new class of model-based metrics by using probabilistic cross-categorization (PCC) (Mansinghka et al., 2016) to learn a generative model of the real tabular data. Leveraging the PCC as a surrogate to the ground-truth data generating process, we form a new spectrum of metrics that readily spans all substructures. The inherent advantage of the model-based technique over the model-free metrics is that all the metrics in this class can be derived using the same underlying assumptions for any substructure and data type. We show in Section 5 that PCC is a decent surrogate.

We demonstrate our structured framework by evaluating eight synthesizers on three datasets. The eight synthesizers include GAN-based (Xu et al., 2019; Patki et al., 2016), autoencoder-based (Xu et al., 2019; Patki et al., 2016), copula-based (Patki et al., 2016), transformer-based (Borisov et al., 2022b), diffusion-based (Kotelnikov et al., 2023), and structure-based (PCC and Nowok et al., 2016) methods. The three datasets span a wide range of data sizes and contain a mix of numeric and categorical columns, some with missing values. The experiments reveal that structure-based synthesizers which explicitly represent the tabular structure (i.e., column distributions and dependencies) often outperform deep-learning synthesizers that use an implicit representation (e.g., via the competition between a generator and discriminator as in GAN).

In summary, the contributions of this paper are:

- A coherent and complete evaluation framework that for the first time allows one to reason about the completeness and coherence of any set of metrics, and unifies existing metrics based on a single, mathematical objective (Section 3).
- An open-source implementation of the evaluation framework[1], featuring a new class of metrics, a comprehensive set of baselines, a principled ordering of metrics that hightlist the utility of the structural perspective, and a tabular data synthesizer (PCC) that allows arbitrary conditioning with missing values (Sections 3.2, 3.3, and 5).
- We demonstrate the structured evaluation framework with a diverse set of 8 synthesizers across 3 datasets of varying sizes, showing that synthesizers with an explicit representation of the tabular structure outperform those without, especially on smaller datasets (Section 5).

## 2 RELATED WORK

**Evaluation.** Many evaluation metrics exist and have been taxonomized largely based on a mix of methodology and structural concerns (Dankar et al., 2022; Afonja et al., 2023; Choi et al., 2017; Zhao et al., 2021). Some work proposes baselines and an aggregate metric to simplify the evaluation (Chundawat et al., 2022). Our work motivates these metrics and baselines from a single unifying objective. See Appendix E for a detailed discussion.

**Tabular data synthesizers.** Approaches can be broadly categorized as: structured parametric, such as Bayesian network models (Zhang et al., 2017; Kaur et al., 2021); structured nonparametric, including probabilistic cross-categorization (Mansinghka et al., 2016) and methods originating from

---

[1]url TBA

the field of statistical disclosure control that models the full joint distributions by a chain-rule decomposition (Nowok et al., 2016); and non-structured over-parametric, encompassing most methods powered by deep learning (Borisov et al., 2022a; Kotelnikov et al., 2023; Kim et al., 2022). One feature we study in this paper is whether the explicit versus implicit representation of the tabular structure affects the synthesizer's performance.

**Tabular data analysis.** Recent works have shown that tree-based methods can outperform deep-learning methods in tabular data analysis (Gorishniy et al., 2021). Grinsztajn et al. (2022) identified three concrete failure modes: susceptibility to uninformative features, bias towards overly smooth function, and negligence of data orientation. The second and third failure modes coincide with known issues for deep-learning tabular data synthesizers, namely, the difficulties in modeling heterogeneous data types and the column dependency structure (Xu et al., 2019; Ma et al., 2020).

## 3 STRUCTURED EVALUATION FRAMEWORK

**The objective.** The goal of a data synthesizer is to produce samples that are statistically analogous to the real data but not direct copies of them. A common assumption for tabular data is that the rows are exchangeable, and thus, the data distribution is fully described by the joint distribution of the columns. Let the columns be $\{c_j \mid j = 1, \cdots, m\}$, the synthetic data distribution be $Q$, the synthetic dataset be $S$, the real data distribution be $P$, and the real dataset be $X$. The objective of the data synthesizer is then $Q = P$ and $S \neq X$. The first part means that joint distributions should match, i.e., $Q(c_1, \cdots, c_m) = P(c_1, \cdots, c_m)$. The second part means that $S$ should be not a direct copy of $X$ but should be *sampled* from $Q$, just as $X$ is a sample from $P$.

**Sketch of analysis.** To derive existing metrics from the main objective, we go through the following general steps:

1. Identify a substructure of $Q$, denoted by $q$, and similarly $p$ from $P$. Consider whether $q = p$ is necessary and sufficient for $Q = P$. The substructures we study include univariate marginal distributions, pairwise joint distributions, leave-one-out conditional distributions, the full-joint distribution, and the structure of missingness.

2. Form an estimate of the substructure from the data; that is, $f_q$ of $q$ from $S$, and $f_p$ of $p$ from $X$. Identify whether $f_q = f_p$ is necessary and sufficient for $p = q$. Each existing metric is related to an estimate. Popular estimates include simple statistics (e.g., mean and support), finite-sample probability mass functions, and distributions represented by ML models.

3. Compute a metric score $t$ of how close $f_q$ and $f_p$ are. We design $t$ to be within the range $[0, 1]$ so that $t = 1$ implies $f_q = f_p$.

4. Repeat steps 2 and 3 for all possible arrangements of the substructure (all combinations of the columns) and then compute the average score.

Going backwards from steps 3 to 1, we form a chain of necessary and/or sufficiency conditions from the metric score, to the estimates, to the substructures, and to the main objective. That is, we analyze whether each relationship of the chain $(t = 1) \Leftrightarrow (f_q = f_p) \Leftrightarrow (q = p) \Leftrightarrow (Q = P)$ holds. Sufficiency is desired, but in most cases only necessary conditions are satisfied. Table 1 summarizes the substructure, estimates, and score of the metrics used in the paper.

### 3.1 THE SPECTRUM OF STRUCTURE

In this subsection we highlight the less obvious connections along the chain of necessary and sufficient condiitons (e.g., that between machine learning efficacy and leave-one-out conditional distribution), and leave the more obvious ones (e.g., with the marginal and pairwise distributions) to Appendix A. We focus on the model-free metrics here and leave the discussion of the model-based (or PCC-based) metrics to Section 3.3.

**Marginal and pairwise distribution.** Equivalence between all univariate marginals is necessary but not sufficient for the equivalence of the full joint distribution: $Q = P \Rightarrow Q(c_j) = P(c_j)$ for $j = 1, \cdots, m$. The same statement holds for pairwise distributions: $Q = P \rightarrow Q(c_j, c_k) = P(c_j, c_k)$ for all $j \neq k$. The metrics used for the marginal substructure include estimates based on simple statistics as well as probabilty mass/density functions. For the pairwise substructure, distributional

| Metric (data type) | Estimate $f_q, f_p$ | Score $t$ | Implication |
|---|---|---|---|
| **Marginal** | | | |
| TVComplement (categorical) | PMF of $S[c_j]$ | $1 - $ TV dist$(f_q, f_p)/Z$ | $t = 1 \Leftrightarrow Q(c_j) = P(c_j)$ as $n \to \infty$ |
| KSComplement (continuous) | CDF of $S[c_j]$ | $1 - $ KS stat$(f_q, f_p)$ | same as above |
| Simple-statstics based (either)✱ | support, range, mean, median, std of $S[c_j]$ | $f_q/f_p$ or $1 - |f_q - f_p|/Z$ | $t = 1 \Leftarrow Q(c_j) = P(c_j)$ |
| PCC-Marginal (both)★ | empirical CDFs from $\hat{P}(S_{i,j})$ | $1 - \frac{1}{2}|ppe|$ | $t = 1 \Leftarrow Q(c_j) = \hat{P}(c_j)$ as $n \to \infty$ |
| **Pairwise** | | | |
| Contingency-Sim (categorical) | PMF of $S[c_j, c_k]$ | $1 - $ total variation dist$(f_q, f_p)/Z$ | $t = 1 \Leftrightarrow Q(c_j, c_k) = P(c_j, c_k)$ as $n \to \infty$ |
| Correlation-Sim (continuous) | $\rho(S[c_j], S[c_k])$ | $1 - |f_q - f_p|/Z$ | $t = 1 \Leftarrow Q(c_j, c_k) = P(c_j, c_k)$ |
| MutualInformation-Sim (both)★ | $MI(S[c_j], S[c_k])$ | $1 - \sum |f_q - f_p|/Z$ | same as above |
| PCC-Pairwise (both)★ | empirical CDFs from $\hat{P}(S_{i,(j,k)})$ | $1 - \frac{1}{2}|ppe|$ | $t = 1 \Leftarrow Q(c_j, c_k) = \hat{P}(c_j, c_k)$ as $n \to \infty$ |
| **Leave-one-out (LOO)** | | | |
| ML efficacy (categorical)✱ | $f_q : ML(S[c_j] \mid S[\{c_{-j}\}]); f_p : X[c_j]$ | acc between argmax $f_q(c_j \mid X[\{c_{-j}\}])$ and $X[c_j]$ | higher $t$ generally implies $Q$ more similar to $P$ |
| Privacy metrics (either)✱ | $f_q : ML(S[\{c_\alpha\}] \mid S[\{c_\beta\}]); f_p : X[\{c_\alpha\}]$ | $1 - a(X[\{c_\alpha\}]; f_q(\{c_\alpha\} \mid X[\{c_\beta\}]))$ | higher $t$ generally implies $Q$ less similar to $P$ |
| PCC-LOO (either)★ | empirical CDFs from $\hat{P}(S_{i,j} \mid S_{i,\{-j\}})$ | $1 - \frac{1}{2}|ppe|$ | $t = 1 \Leftarrow Q = \hat{P}$ as $n \to \infty$ |
| **Full joint** | | | |
| ML-Detection (both)✱ | discriminator $f$ of $Q(r)$ trained on $S$ | $2(1 - ROC\ AUC)$ | $t = 1 \Leftarrow Q = P$ as $n \to \infty$ |
| PCC-FullJoint (both)★ | empirical CDFs from $\hat{P}(S_i)$ | $1 - \frac{1}{2}|ppe|$ | $t = 1 \Leftarrow Q = \hat{P}$ as $n \to \infty$ |
| **Missingness** | | | |
| MissingValue-Sim (either) | fraction of missing values in $S_\nu[c_j]$ | $1 - |f_q - f_p|$ | $t = 1 \Leftrightarrow Q_\nu(c_j) = P_\nu(c_j)$ as $n \to \infty$ |
| Missing-NotAtRandom-Sim (both)★ | correlation matrix formed from column pairs in $S_\nu[\{c_\nu\}]$ | $1 - \sum |f_q - f_p|/Z$ | $t = 1 \Leftarrow Q_\nu(\{c_\nu\}) = P_\nu(\{c_\nu\})$ |
| Covariate-DependentMissing-Sim (both)★ | correlation matrix formed by correlating columns in $S_\nu[\{c_\nu\}]$ with those in $S_\nu[\{c_{-\nu}\}]$ | $1 - \sum |f_q - f_p|/Z$ | $t = 1 \Leftarrow Q_\nu(\{c_\nu\} \mid \{c_{-\nu}\}) = P_\nu(\{c_\nu\} \mid \{c_{-\nu}\})$ |

Table 1: A summary of the evaluation metrics in the structured evaluation framework. See main text and Appendix A for description. Code is at `TBA`. In the Estimate column, we summarize the construction of $f_q$ from the synthetic dataset $S$; $f_p$ is contructed from the real dataset $X$ in the same way but left out for brevity, unless otherwise noted. A column $c$ of a distribution is denoted by $Q(c)$ in paranthesis, while a column $c$ of a dataset is denoted by $S[c]$ in brackets. The symbol ★ denotes newly introduced metrics, ✱ means the entry corresponds to multiple metrics.

properties are estimated by contingency tables, correlation coefficients, and mutual information. For the full chain of connection from metrics scores to substructures, see Appendices A.1–A.2.

**Leave-one-out conditional (LOO) distribution.** The full joint distributions are equal if and only if all the *leave-one-out conditional (LOO) distributions* are equal: $Q = P \Leftrightarrow Q(c_j \mid \{c_{-j}\}) = P(c_j \mid \{c_{-j}\})$ for $j = 1, \cdots, m$, where $\{c_{-j}\}$ denotes all columns except the $j^{th}$ one. Necessity holds because the LOO conditional is a subset of the full joint. Sufficiency holds because the complete set of the LOO conditionals fully specifies the complete factorization of the full joint under the chain rule. Mathematically, the chain-rule factorization of the full joint is: $Q = Q(c_m \mid c_1, \cdots, c_{m-1})Q(c_{m-1} \mid c_1, \cdots, c_{m-2}) \cdots Q(c_2 \mid c_1)Q(c_1)$. Then observe that for each target column (the LHS of a conditional), the conditions (the RHS of the conditional) of the factor are always a subset of the conditions in the LOO conditional distribution, i.e., $\{c_1, \cdots, c_{j-1}\} \subseteq \{c_{-j}\}$. Thus, given the same target column, the specification of the LOO conditional also fully specifies the corresponding factor in the chain-rule expression.

The popular metric of machine learning (ML) efficacy belongs to this substructure.[2] The idea of ML efficacy is that an ML algorithm trained on the synthetic data should perform as well as one trained on the real data. By identifying the training features or regressors as the conditions on the LHS of the LOO conditional, and the training labels or response variable as the target column on the RHS of the LOO conditional, we see that the trained ML is an estimate $f_q$ of the LOO distribution: $Q(c_j \mid \{c_{-j}\})$. The estimate $f_q$ approaches the distribution as $n_{train} \to \infty$ if the ML model is a universal approximator and the training algorithm provably converges to the global minimum. The estimate $f_p$ of the real LOO distribution is replaced by the real dataset, which consists of samples from the distribution. The score takes the form of the accuracy between the real label and the predictions of the ML model, which is trained on the synthetic data and tested on the real data. Mathematically, $t(f_q) = a(X[c_j], \mathrm{argmax} f_q(c_j \mid X[\{c_{-j}\}]))$, where $a(\cdot)$ denotes an accuracy function, $X[c_j]$ the real labels, and $f_q(c_j \mid X[\{c_{-j}\}])$ the ML model's prediction given the test features/regressors. Note that because $f_p$ is a set of samples, the argmax prediction may not attain maximum score even if $Q = P$; thus, neither sufficiency nor necessity hold. See Appendix A.3 for the implementation details, how the SELF baseline promote necessity, and a variant metric called column-wise prediction (Choi et al., 2017; Engelmann & Lessmann, 2021).

**The full joint.** The likelihood of any row of data is the same under the two models if and only if the full joint distributions are equal: $Q = P \Leftrightarrow Q(r) = P(r)$ for any row of data $r$. Instead of tackling the tall task of explicitly estimating $Q(r)$ and $P(r)$, one way to get at $Q(r) = P(r)$ is by learning a discriminator $f$ between $Q(r)$ and $P(r)$ from $S$ and $X$, respectively. Metrics that learn an ML model to discriminate between rows of the real data from rows of the synthetic data fall under this category. A perfect $f$ (resulting from a poor synthesizer) would have an $ROC\ AUC$ of 1, and a random $f$ (resulting from $Q = P$) would have an $ROC AUC$ of $1/2$. Thus, the score is $t = 2(1 - ROC\ AUC)$. As $n_{test} \to \infty$, $t = 1$ is necessary for $Q = P$. Sufficiency likely does not hold because there may be multiple ways that the implicit $f_q$ and $f_p$ can produce a discriminator $f$ to achieve perfect score. See Appendix A.4 for implementation details.

**Missingness.** Consider the missingness of each entry of a table as a binary variable. We can then construct a binary table of missingness $S_\nu$ from the synthetic dataset $S$. Just as $S$ is a sample from $Q$, we say $S_\nu$ is a sample from $Q_\nu$. Since missingness is a property of the data generating process, $Q_\nu = P_\nu$ is necessary but not sufficient for $Q = P$. In Appendix A.5, we consider three substructures of the objective $Q_\nu = P_\nu$. (1) The marginals are equivalent: $Q_\nu(c_j) = P_\nu(c_j)$ for all $j$. (2) The joint distributions of columns with missing values are equivalent: $Q_\nu(\{c_j\}) = P_\nu(\{c_j\})$, where $\{c_j\}$ is the set of columns with missing values. This leads to a novel metrics related to missing-not-at-random. (3) The distributions of missing values conditioned on non-missing values are equivalent: $Q_\nu(\{c_j\} \mid \{c_k\}) = P_\nu(\{c_j\} \mid \{c_k\})$, where $\{c_k\}$ are the set of columns without any missing value. This leads to another novel metric related to covariate-dependent missingness. Note that all three conditions above are necessary but generally not sufficient for $Q_\nu = P_\nu$.

**Privacy.** While we would like to have $Q = P$, we do not want the synthesizer to simply memorize the training data, resulting in $S = X$. Metrics on privacy against inference punishes memorization. Given sensitive columns $\{c_\alpha\}$ and insensitive columns $\{c_\beta\}$, a privacy attacker is trained on the synthetic data and predicts the sensitive columns in the real data given the insensitive columns. If

---

[2]Our use of ML efficacy is a slight generalization of its usual version in that we loop through all the columns.

$S = X$, the attacker can simply find the row in the synthetic data with insensitive values cloest to the insensitive values in the real data, and predict the real sensitive value by reading off the sensitive value in the synthetic data. Privacy attack can be expressed in terms of conditional distributions: the attacker learns an estimate $f_q$ of $Q(\{c_\alpha\} \mid \{c_\beta\})$, and then use $f_q$ to predict the sensitive columns of $X$ conditioned on the insensitive columns of $X$, akin to the estimate $f_q$ in ML efficacy. The privacy-against-inference score is $t = 1 - a(X[\{c_\alpha\}], f_q(\{c_\alpha\} \mid X[\{c_\beta\}]))$, where $a(\cdot)$ is an accuracy function. From this perspective, it is clear that the ML efficacy and privacy against inference metrics are anti-correlated, posing a tradeoff between learning the real distribution $P$ and guarding against inference of $P$ (Figure S4). See Appendix A.6 for implementation details.

## 3.2 BASELINES

In this subsection we introduce baselines that provide empirical upper and lower bounds on the metrics by using the real data as the synthetic data. The first baseline, SELF, motivated by the fidelity part of the objective $Q = P$, simply use a direct copy of the real data as the synthetic data, or $S = X$. This baseline provides an upper bound for all the metrics, except for the privacy metrics, for which it provides a lower bound. The substructure of marginal distribution motivates the PERM baseline: the synthetic data is constructed by permuting each column of $X$ independently. This baseline preserves a high score for metrics in the marginal group and provides a reasonable *lower bound* for the metrics related to higher-order substructures. A data synthesizer that learns any form of column dependency should outperform the PERM baseline. The full objective $Q = P$ and $S \neq X$ motivates the HALF baseline, where the real data is split in half, with one half representing $X$, and the other half representing $S$. Such splitting ensures that both $S$ and $X$ are sampled from $P$. Aside from variations due to smaller sample size, this baseline provides realistic target values for *all* the metrics and represents the performance of a good data synthesizer.

## 3.3 MODEL-BASED / SURROGATE METRICS

The idea of this class of metrics is to learn a surrogate model of the true data-generating distribution. This surrogate model should have the property of arbitrary conditioning, i.e., the ability to output the probability of any set of columns given any non-overlapping set, including the empty set. This property allows direct evaluation of any targeted substructure. Furthermore, unlike the model-free metrics which requires different estimators and scores for different datat type and structural properties as exemplified in Table 1, this class of mode-based metrics can use the same estimator and score for all data type and substructure, as demonstrated below.

There are multiple ways to make use of such a surrogate model for evaluation. Here, we present a simple way that involves only the surrogate model of the original data (but not of the synthetic data), the original dataset $X$, and the synthetic dataset $S$. We begin the description with the full joint distribution, which then readily extends to all substructures. First compute the likelihood of the real data samples under the surrogate model one row at a time, forming a set of likelihoods, denoted by $\{\hat{P}(X_i) \mid i = 1, \cdots, n\}$, where $X_i$ is the $i^{\text{th}}$ row of $X$. Compute the same quantity from the synthetic dataset to obtain $\{\hat{P}(S_i) \mid i = 1, \cdots, n\}$. Construct an empirical CDF from each list, forming $f_q$ and $f_p$. Plot the two CDFs against each other as in a probability-probability plot. If the two CDFs are equal, the plot will coincide with the $x = y$ line. Thus, the metric socre is one minus the deviation from the $x = y$ line. More precisely, $t = 1 - \frac{1}{2}|ppe|$, where $ppe$ is the difference between the pp-plot and the line $y = x$ for $x = [0, 1]$. Maximum $t$ is necessary and sufficient for the equivalence of the two empirical CDFs by design: $t = 1 \Leftrightarrow f_q = f_p$. Given that the surrogate model is not a uniform distriubtion over the variables, as $n \to \infty$, the equivalence of the CDFs is necessary but not sufficient for the equivalence of the synthetic and real distribution as modeled by the surrogate: $f_q = f_p \Leftarrow Q = \hat{P}$. Sufficiency does not hold because there may be multiple ways to obtain the same empirical CDFs. The extension to a substructure is done by replacing a full row with certain entries of a row conditioned on other entries of the same row.

For the surrogate model, we used an optimized version of probabilistic cross-categorization (PCC) (Mansinghka et al., 2016). PCC is a Bayesian nonparametric model that learns the distributional property of each column as well as the dependencies among the columns and rows. This structured representation of tabular data allows us to extract conditional distributions of any set of columns given any non-overlapping set, including the empty set. Below we give a brief description of PCC.

**Probabilistic cross-categorization.** Probabilistic Cross-Categorization (PCC; Mansinghka et al. (2016)) is a hierarchical Bayesian nonparametric architecture designed for tabular data. Given a table with $n$ rows and $m$ features, PCC uses a Dirichlet process mixture model (Antoniak, 1974) to cluster features into between 1 and $m$ *views*, and within each view, uses another Dirichlet process to cluster rows into between 1 and $n$ *categories*. To allow features to take on different types, including missing values, features are modeled independently.

The generative process is as follows: the Chinese Restaurant Process (CRP Aldous (1985)) discount parameter, $\alpha$, is drawn from a Gamma distribution for both the view assignment and the category assignments. The view assignment, $v$, is drawn from $CRP(\alpha_v; m)$. For each view, the assignment of rows to categories within that view is drawn from $CRP(\alpha_{c_u}; n)$. A prior distribution, $\phi_j$ for each column, $j$, is drawn from a hyper prior, $H_j$. The mixture distribution component model parameters for category $k$ in column $j$, $\theta_{k,j}$ is drawn from $\phi_j$. To summarize:

$$\alpha_v \sim Gamma(1,1) \qquad v \sim CRP(\alpha_v; m) \qquad \phi_j \sim H_j$$
$$\alpha_{c_1}, \ldots, \alpha_{c_{|v|}} \sim Gamma(1,1) \qquad c_u \sim CRP(\alpha_{c_u}; n) \qquad \theta_{k,j} \sim p(\theta|\phi_j)$$
$$x_{i,j} \sim p(x|\theta_{c_{v_j,i},j})$$

where $v$ is an $m$-length vector with $v_j$ being the index of the view to which column $j$ is assigned; $c_u$ is an $n$-length vector with $c_{u,i}$ being the index of the category to which row $i$ is assigned under view $u$; $H_j$ is the hyper prior on column $j$; $\phi_j$ is the prior on column $j$; $\theta_{k,j}$ are the parameters of the $k^{th}$ component of the mixture distribution for column $j$; and $x_{i,j}$ is the datum at cell $(i,j)$. The joint distribution is

$$p(\alpha_v)p(v|\alpha_v) \prod_{u=1}^{|v|} \left[ p(\alpha_u)p(c_u|\alpha_u) \prod_{\{j;v_j=u\}} \left[ p(\theta_j|\phi_j)p(\phi_j|H_j) \prod_{i=1}^{n} p(x_{ij}|\theta_{c_{u,i},j}) \right] \right]. \quad (1)$$

See Appendix B for implementation details.

## 4 EXPERIMENT

**Synthesizers.** We evaluate 8 methods: PCC, synthpop (Nowok et al., 2016), DDPM (Kotelnikov et al., 2023), GReaT (Borisov et al., 2022b), GaussianCopula, TVAE (Xu et al., 2019), CTGAN (Xu et al., 2019), and CopulaGAN. Synthpop and PCC model tabular structure explicitly. DDMP is based on deep diffusion models, and GReaT on large language models, TVAE on variational autoencoders, and CTGAN and CopulaGAN on generative adversarial networks. GaussianCopula is a copula model that captures pairwise statistics. We also provide three model-free baselines: SELF, PERM, and HALF (Section 3.2). See Appendix C for implementation details.

**Datasets.** The evaluation is run on three datasets curated in SDV.[3] We select datasets that contain missing values and have a good mix of categorical and numeric columns. The three datasets used are: `student` (215 rows, 7 categorical columns, 7 numeric columns, 2 time-stamp columns, and 4 columns with missing value), `expedia` (1000 rows, 16 categorical columns, 5 numeric columns, 3 time-stamp columns, and 1 column with missing value), and `census` (299285 rows, 29 categorical columns, 12 numeric columns, and 1 column with missing value).

**Metrics.** We gathered a total of 32 evaluation metrics: 9 are marginal-based; 4 are pairwise-based; 7 are based on the leave-one-out conditional; 3 are full-joint-based; 4 concerns missingness; and 5 relates to privacy. Out of the 32, we introduced and implemented 7 novel metrics. The remaining 25 are implemented in SDMetrics (Dat, 2022). See Appendix A for full details.

The evaluation procedures are as follows: (1) Train the synthesizer models on the original data. (2) For each synthesizer and baselines, generate a synthetic dataset. (3) Compute the 32 metrics given the synthetic dataset and real dataset. Computation of the metrics is always done on all possible combinations of column subsets. For example, for the pairwise-based metrics, we compute a score for every column pair; for the metrics based on leave-one-out conditionals, we loop through all columns as the target column. (4) Repeat steps 2 and 3 five times, each time with a newly synthesized dataset. The code to reproduce the results is at TBA.

---

[3]Downloaded from `http://sdv-datasets.s3.amazonaws.com/index.html`

## 5 EVALUATION RESULTS

**Natural ordering of the metrics.** The structured framework assigns a natural ordering to the metrics based on the complexity of the substructure, from univariate marginal, to pairwise joint, to leave-one-out (LOO) conditionals, and to the full joint distribution. We position the missingness group before the marginal group because missingness of the table translates to a binary table, which is structurally much less complex than the original table. Figure 2 shows that the scores generally decrease as the complexity of the substructure increases. This is expected because there is less statistical information to be learned from a simpler substructure (e.g., univariate marginal) than from a more complex one (e.g., the leave-one-out distribution).

**Validation of PCC-based metrics.** The main validation criterion is that the PCC-based metrics exhibit the same trend as the model-based metrics. Figures S1–S3 confirm that the orderings of the different synthesizer and baselines are similar under the PCC-based and model-free metrics. The second criterion is that PCC is a good surrogate model of the data generating process. This is confirmed by the PCC being among the highest performing methods according to the model-free metrics. Note that while the synthpop may be a better surrogate model, it does not allow arbitrary conditioning nor computes the likelihoods like PCC does (see Section 3.3); thus, it cannot be used to generate the model-based metrics described. Furthermore, PCC-based evaluation is more robust, as evidenced by its smoother trend and smaller error bars relative to model-free evaluation (Figures S1–S3). In particular, the robustness helps stabilize the baselines. Once a PCC model is trained on the original data, the PCC-based metrics are also faster to compute than the model-free ones.

**Comparison of synthesizers.** Figure 2 shows the structured evaluation of the synthesizers and baselines across the 3 datasets. We first observe that all the methods exhibit the general trend of score decreasing as a function of substructure complexity; that is, the non-monotonicity is small compared to the overall decrease. We also observe that the spread of synthesizers' scores increase as the complexity of the substructure increases, allowing one to differentiate the performance of the methods. These observations confirm that this structural perspective is useful for evaluation.

For the `student` dataset, synthpop, PCC, DDPM, and GaussianCopula form a group of top performers. Using the baselines, we identify that the data generated from CTGAN and CopulaGAN are similar to the PERM version of the real data, meaning that they captured the column-wise statistics but not the column dependencies. For the `expedia` dataset, synthpop is the clear winner, with PCC, TVAE, and GReaT as runner-ups. Note that GReaT has the lowest missingness score because it generates missing values in columns that do not have any missing values in the original data. DDPM is the worst performer here because it only generates one value for each categorical column (see Appendix C for training details). For the larger `census` data, all methods evaluated, except for GaussianCopula, reach a good performance, while synthpop is still the best. Overall, we see that synthesizers which model structure explicitly (synthpop and PCC) performs well consistently, even on smaller datasets. On the contrary, methods that do not explicitly make use of the tabular structure (especially CTGAN and CopulaGAN) require more data and careful optimization to reach similar performance. The performance of GaussianCopula decreases as data size increased likely because the method is not desinged to capture the full joint. See Figure S5 for correlations among the metric groups.

To quantify the performance, we compute the *quality* score of each synthesizer. The HALF baseline is the closest model to the main objective of sampling from a distribution $Q$ that is equal to $P$. Thus, we compute the quality score by taking the absolute difference between a synthesizer's score trace and that of the HALF baseline, then averaging over the substructures. The quality scores are presented in Table 2, along with the training and sampling time of each method. All times are wall clock time in seconds as recorded on a 2021 MacBook Pro with the 32GB Apple M1 Pro chip.

**Practical utility.** The structured evaluation shows the degree and extent to which the full joint is mimicked, allowing users to select synthesizer with confidence. The metric groups with the baselines show where along the structural spectrum a data synthesizer falls short, signaling where developers can improve the synthesizer. The framework presents evaluation results in a meaningful, coherent ordering to help human evalutors identify strange behaviors more easily. The structured spectrum, the analysis of the necessity and sufficiency chain, and the fluctuations of the metric values along the spectrum can aid the design of new metrics (e.g., 3-way interactions and leave-n-out), improvement of the metric estimators, and identification of metric shortcomings, respectively.

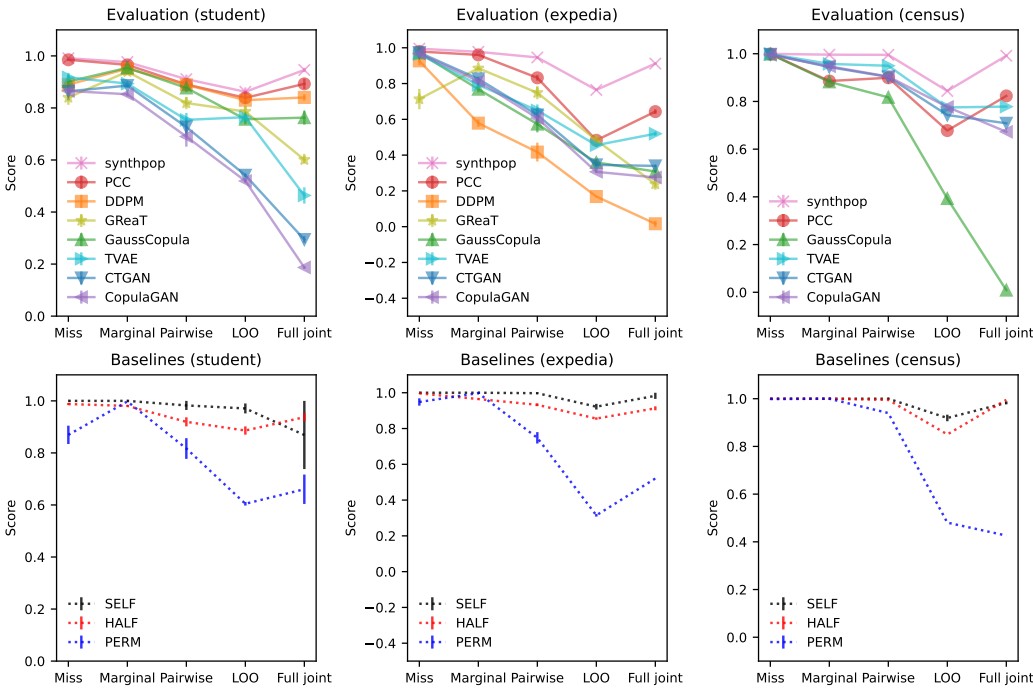

Figure 2: **Structured evaluation on datasets of different sizes.** The scores shown are the model-free and PCC-based scores. The error bars are combined assuming independence. For the `census` data, we omitted DDPM and GReaT for their poor quality and computational cost, respectively.

| Model | student | | expedia | | census | |
|---|---|---|---|---|---|---|
| | Qual | Train + Sample | Qual | Train + Sample | Qual | Train + Sample |
| synthpop | 0.99 | 0.097 | 0.98 | 0.48 | 0.99 | 35.97 |
| PCC | 0.97 | 18.00 + 0.0011 | 0.85 | 26.97 + 0.013 | 0.89 | 4446 + 5 |
| DDPM | 0.94 | 464.73 + 5.93 | 0.49 | 1397 + 50 | — | 2270 + 17544 |
| GReaT | 0.85 | 1262 + 37 | 0.68 | 14099 + 1819 | — | >90000 |
| GaussianCopula | 0.91 | 0.11 + 0.021 | 0.66 | 0.33 + 0.082 | 0.65 | 25.94 + 12.01 |
| TVAE | 0.82 | 2.19 + 0.039 | 0.74 | 25.57 + 0.092 | 0.92 | 4213 + 10 |
| CTGAN | 0.72 | 13.37 + 0.044 | 0.69 | 146.21 + 0.14 | 0.89 | 14756 + 16 |
| CopulaGAN | 0.68 | 13.08 + 0.056 | 0.66 | 147.21 + 0.17 | 0.89 | 15618 + 19 |

Table 2: **Synthesizer quality and speed**. "Sample" refers to the time to sample a dataset of the same size as the training set. For the `census` data, we omitted DDPM and GReaT because of poor quality and computational cost, respectively.

## 6  CONCLUSION

We describe a framework that evaluates the degree to which tabular data synthesizers produce samples from the real data distribution via a structural lens. The framework unifies existing metrics, motivates baselines, inspires novel metrics, and offers the most coherent and complete evaluation to date. Applying the evaluation to a variety of synthesizers, we observe that the explicit representation of the tabular structure is advantageous. Limitations of the paper and future works include using the structured framework (1) to improve model-free metrics, (2) to study differential privacy, and (3) to understand the adverse effects of augmenting ML training with synthetic data (Shumailov et al., 2023; Alemohammad et al., 2023).

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

# A METRICS

All metrics implemented by SDMetrics are documented at `https://docs.sdv.dev/sdmetrics/metrics/metrics-glossary`. We follow the naming convention in SDMetrics' documentation. The metric scores for all the metrics range between $[0, 1]$.

## A.1 MARGINAL

We use 9 metrics for the marginal substructure. Two are for categorical columns: **CategoryCoverage**, **TVComplement**; six are for numeric columns: **BoundaryAdherence**, **KSComplement**, **MeanSimilarity**, **MedianSimilarity**, **RangeCoverage**, **StdSimilarity**; and **PCC-Marginal** is for either. All these metrics are implemented by SDMetrics, except for PCCMarginal. Each categorical (numeric) metric is run on every categorical (numeric) column in the dataset and then averaged.

The estimates $f_q$ computed on column $j$ of $S$ is denoted by $S[c_j]$, and $f_p$ is the estimate computed on $X[c_j]$. Among the above metrics, the estimate used in TVComplement is the finite-sample probability mass functions (PMF), and the estimate used in KSComplement is finite-sample cumulative distribution functions (CDF). As the number of samples $n$ approaches infinity, $f_q = f_p$ is necessary and sufficient for $Q(c_j) = P(c_j)$, because the estimates approach the target marginals. A typical form of the score is $t = 1 - dist(f_q, f_p)/Z$, where $dist(\cdot)$ is some distance function measuring the difference between the input sampling distributions. Distance functions used here are the Kolmogorov–Smirnov statistic and the total variation distance for CDF and PMF, respectively. Since both distances are zero only then the distributions being compared are equal, $t = 1$ is sufficient and necessary for $f_q = f_p$.

As described in the main text, for PCC-Marginal, the estimate $f_q$ is the empirical CDF constructed from the probability of row $i$ and column $j$ under the PCC model for each row in the synthetic dataset, or mathematically, $\{\hat{P}(S_{i,j}) \mid i = 1, \cdots, n\}$. Similarly, the estimate $f_p$ is the empirical CDF constructed on the real dataset $X$: $\{\hat{P}(X_{i,j}) \mid i = 1, \cdots, n\}$. The score is $t = 1 - \frac{1}{2}|ppe|$, where $|ppe|$ is area of the absolute difference between the pp-plot and the line $y = x$ in the domain $x = [0, 1]$. The pp-plot is the plot constructed from $(x = f_q, y = f_p)$. By design, $t = 1 \Leftrightarrow f_q = f_p$. As the number of sample approach infinity, $f_q = f_p \Leftarrow Q = \hat{P}$ because two equal distributions would generate the same CDF.

The rest of the metrics use simple statistics as the estimates – support, range, mean, median, and standard deviation – of each column. Because these statistics are not sufficient statistics, $f_q = f_p$ is necessary but not sufficient for $Q(c_j) = P(c_j)$. For these estimates, the score typically takes the form of $t = 1 - |f_q - f_p|/Z(X[c_j])$, where $Z(X[c_j])$ is an appropriate normalizer computed on $X[c_j]$ such that $t \in [0, 1]$. Here, $t = 1$ is necessary and sufficent for $f_q = f_p$, since the mapping between the score and the absolute difference of the two estiamtes is one-to-one.

## A.2 PAIRWISE

We use 4 metrics for the pairwise substructure: **ContingencySimilarity** for categorical column pairs, **CorrelationSimilarity** for numeric column pairs, and **MutualInformationSimilarity** and **PCC-Pairwise** for both types. The first two metrics are implemented by SDMetrics. Note that metrics based on pairwise distributions cover more structure than the ones above, but do not speak to multi-way interactions.

For ContingencySimilarity, the estimate for $f_q$ is the finite-sample PMF formed from a pair of columns from the synthetic dataset, $S[c_j, c_k]$. Similarly, $f_p$ is formed from $X[c_j, c_k]$. These PMF estimates are contingency tables, and $f_q = f_p$ is necessary and sufficient for $Q(c_j, c_k) = P(c_j, c_k)$ as $n \to \infty$. The score is one minus a normalized total variation distance between the PMFs. Maximum score is necessary and sufficient for $f_q = f_p$ because the total variation distance have a one-to-one relationship with the metric score.

For CorrelationSimilarity, the estimate $f_q$ is the correlation coefficient between two columns of the synthetic data, denoted by $\rho(S[c_j], S[c_k])$. Similarly, $f_p = \rho(X[c_j], X[c_k])$. Because there are infinite ways to arrive at the same correlation coefficient, $f_q = f_p$ is necessary but not sufficient for $Q(c_j, c_k) = P(c_j, c_k)$. The score is $t = 1 - |f_q - f_p|/Z$. Maximum score is necessary and

sufficient for $f_q = f_p$ because the absolute difference of the two estiamtes is one-to-one with the metric score by desgin. Note that limitations of correlation are less in play here: we expect roughly linear relationships between column pairs, although outliers could still bias the score.

A shortcoming of the implementation of ContingencySimilarity and CorrelationSimilarity is that they do not apply when the two columns are of different data types. To fill this gap, we propose a novel model-free metric: MutualInformationSimilarity. The estimate $f_q$ is the mutual information between any pair of columns in the synthetic data, i.e., $MI(S[c_j], S[c_k])$. Similarly, $f_p = MI(X[c_j], X[c_k])$. The mutual information is computed using sklearn (Pedregosa et al., 2011). Because of the multiplicity of ways to achieve the same mutual information, $f_q = f_p$ is necessary but not sufficient for $Q(c_j, c_k) = P(c_j, c_k)$. The score is constructed as follows: First, construct a mutual information matrix, where the entry at position $[j, k]$ is assigned the value $MI(D[c_j], D[c_k])$, for $D = S$ or $X$. Then, zero the main diagonal and normalize the matrix so that it sums to one to produce matrices $\mathbf{M}_S$ and $\mathbf{M}_X$. Finally, the score is $t = 1 - \sum_{j,k} |\mathbf{M}_S[j, k] - \mathbf{M}_X[j, k]|/2$, which ranges from $[0, 1]$. This score does not require the third general step of averaging, as the average is in the sum already. Maximum score is necessary and sufficent $f_q = f_p$ for the same reason mentioned in the paragraph above.

A further description of the implementation of MutualInformationSimilarity is as follows: If both columns are categorical, we use the mutual_info_classif function in the scikit-learn package to estimate the mutual information; if both columns are numeric, we use the mutual_info_regression function; if the column pair is mixed in type, we take the average of the two functions. Computing the mutual information of every column pair, we obtain two mutual information matrices, one for the real data and one for the synthetic data. The metric score is the sum of the absolute element-wise difference between these two matrices, normalized to range between $[0, 1]$. To restrict computational cost, if the dataset contains more than 10000 rows, we randomly sample 10000 rows from it. Pilot experiments show that, as a function of row number, the change in MutualInformationSimilarity begins to diminish around this dataset size.

The implementation of PCC-Pairwise is as follows: The estimate $f_q$ is the empirical CDF constructed from the probability of row $i$ and any column pair $(j, k)$ under the PCC model for each row in the synthetic dataset, or mathematically, $\{\hat{P}(S_{i,(j,k)}) \mid i = 1, \cdots, n\}$. Similarly, the estimate $f_p$ is the empirical CDF constructed from $\{\hat{P}(X_{i,(j,k)}) \mid i = 1, \cdots, n\}$, which is on the real dataset $X$. The rest of the steps are the same as described in PCC-Marginal. The reasoning for the chain of necessity is also as described in PCC-Marginal. For this metric, if the number of rows of the dataset exceeds 10000, we randomly sample 10000 rows for the evaluation.

### A.3 LOO

We use 7 metrics for the leave-one-out conditional (LOO) substructure. Six for binary/categorical columns: **BinaryAdaBoostClassifier**, **BinaryDecisionTreeClassifier**, **BinaryLogisticRegression**, **BinaryMLPClassifier**, **MulticlassDecisionTreeClassifier**, **MulticlassMLPClassifier**, and one for either: **PCC-LOO** Except for PCC-LOO, all other metrics are implemented by SDMetrics.

We perform three types of preprocessing to ensure that the ML efficacy metrics would run smoothly. First, we ensure that the set of unique values in the train set (synthetic data) is exactly the same as the set of unique values in the test set (real data) by removing values that are outside the intersection of the two sets. Second, to control for computational cost, we restrict both the train and test sets to be at most 15000 rows. Pilot experiments show that the metrics stabilize around this size. For datasets exceeding 15000 rows, we use stratified sampling to ensure that every unique value in the data set is covered. Lastly, we impute all missing values by replacing them with random samples of non-missing values from their respective columns. Given a metric for a data type, the target column is a column of that type in the real dataset, and the feature columns are all the remaining categorical and numeric columns in the synthetic dataset. Each metric is repeated on each admissible target column and associated feature columns, then averaged.

The implementation of PCC-LOO is as follows: The estimate $f_q$ is the empirical CDF constructed from the conditional probability of row $i$ and column $j$, given row $i$ and all columns except for $j$, under the PCC model for each row in the synthetic dataset. Mathematically, $f_q = \{\hat{P}(S_{i,j} \mid S_{i,\{-j\}}) \mid i = 1, \cdots, n\}$. In like manner, the estimate $f_p$ is the empirical CDF constructed from the

real dataset: $\{\hat{P}(X_{i,j} \mid X_{i,\{-j\}}) \mid i = 1, \cdots, n\}$. The rest of the steps and the chain of necessity are as described in PCC-Marginal. For this metric, if the number of rows of the dataset exceeds 10000, we randomly sample 10000 rows for the evaluation.

Following the same steps of computing ML efficacy as outlined in the main text, we see that the SELF baseline provides an estimate $f_p$ by training the ML model on the real dataset. Note that even as $n_{train} \to \infty$, the SELF baseline score $t(f_p)$ does not guarantee $t$ is capped because the argmax prediction may not always match the sample. However, in the large training data limit, $t(f_p)$ does provide the attainable upper bound, if the ML is a universal approximator and the training attains to the global minimum. Thus, as $n_{test} \to \infty$, $t(f_q) = t(f_p)$ is necessary for $Q(c_j \mid \{c_{-j}\}) = P(c_j \mid \{c_{-j}\})$. Note that this comparison is a variant of the general steps outlined in the sketch of analysis, as the comparison is between scores rather than estimates.

Column-wise prediction as presented and used in Choi et al. (2017) and Engelmann & Lessmann (2021) is a close variate to ML efficacy. The difference between dimension-wise prediction and ML efficacy is the following: Let the ML model trained on the synthetic data be $f_q$, and the ML model trained on the real data be $f_p$. In column-wise prediction, the accuracy is computed between the predictions of $f_q$ and $f_p$, whereas in ML efficacy the accuracy is computed between the predictions of $f_q$ and the corresponding column of the real data. Also, ML efficacy does not require splitting the data into a train and a test set, while the dimension-wise prediction uses a split. The chain of necessity holds for the same reason as that a comparison between $f_q$ and the SELF baseline promotes necessity.

## A.4 FULL JOINT

We use 3 metrics for the full-joint (sub)structure: **LogisticDetection**, **SVCDetection**, and **PCC-FullJoint**. The former two are implemented by SDMetrics. To control for their computational cost, we use stratified sampling to sample roughly 4500 rows from each of the real and synthetic datasets. Pilot experiments show that the change in metric values begin to diminish above this dataset size. We also ensure that the real dataset and synthetic dataset have exactly the same set of unique values.

The implementation of PCC-FullJoint is as follows: The estimate $f_q$ is the empirical CDF constructed from the probability of the entire row $i$ under the PCC model for each row in the synthetic dataset: $f_q = \{\hat{P}(S_i \mid i = 1, \cdots, n\}$. Similarly, the estimate $f_p$ is the empirical CDF constructed from real dataset: $f_p = \{\hat{P}(X_i) \mid i = 1, \cdots, n\}$. The rest of the steps and the chain of necessity are the same as described in PCCMarginal.

## A.5 MISSING

We use 4 metrics for the missingness substructure: **MissingValueSimilarityCat**, **MissingValueSimilarityNum**, **MissingNotAtRandomSimilarity**, and **CovariateDependentMissingSimilarity**. The first two are implemented by SDMetrics. The latter two are novel model-free metrics.

The MissingValueSimilarityCat and MissingValueSimilarityNum metrics concern the univariate marginals of the missing distribution: $Q = P \Rightarrow Q_\nu(c_j) = P_\nu(c_j)$ for all $j$. The estimate $f_q$ is simply the fraction of missing values in $S[c_j]$, and similarly for $f_p$. As $n \to \infty$, $f_q = f_p \Leftrightarrow Q_\nu(c_j) = P_\nu(c_j)$. The score is $t = 1 - |f_q - f_p|$. Maximum score is necessary and sufficent for $f_q = f_p$ because the absolute difference and the score have a one-to-one relationship. In words, these two metrics compute the fraction of missing values in a column and report how closely this fraction matches between the real and synthetic datasets. The two metrics are applied to every categorical and numeric column and then averaged.

The MissingNotAtRandomSimilarity metric concerns only columns with missing values $\{c_\nu\}$ and the joint distribution over them: $Q = P \Rightarrow Q_\nu(\{c_\nu\}) = P_\nu(\{c_\nu\})$. This joint distribution fully specifies how data are missing not at random, i.e., how missing values in a column depends on missing values in other columns. The estimate $f_q$ is a correlation matrix with each entry being a correlation coefficient between a column pair of the binary missingness matrix constrcuted from the synthetic data set, using only columns with missing values. This part of the missingness matrix is denoted by $S_\nu[\{c_\nu\}]$. The same goes for $f_p$. The equivalence $f_q = f_p$ is necessary but not sufficient for $Q_\nu(\{c_\nu\}) = P_\nu(\{c_\nu\})$. The score is $t = 1 - \sum |f_q - f_p|/Z$, where the $\sum$ is over the elements

of the matrix, the difference is element-wise, and $Z$ is a normalizer to make $t \in [0, 1]$. Maximum score is necessary and suffcient for $f_q = f_p$ because the absolute difference and the score have a one-to-one relationship. The implementation of this metric is as follows: We compute the Pearson correlation coefficient between all pairings of columns with missing values. These computations form two matrices, and the metric score is the sum of the absolute element-wise difference between these two matrices, normalized to range between $[0, 1]$.

The CovariateDependentMissingSimilarity metric concerns the missing columns conditioned on the non-missing columns: $Q = P \Rightarrow Q_\nu(\{c_\nu\} \mid \{c_{-\nu}\}) = P_\nu(\{c_\nu\} \mid \{c_{-\nu}\})$, where $\{c_{-\nu}\}$ denotes the set of columns that do not have any missing value. This conditional distribution describes how the missing value depends on non-missing values. For this novel metric, the estimate $f_q$ is also the correlation matrix, formed by correlating columns of the missingness matrix that have missing values in the actual matrix $S_\nu[\{c_\nu\}]$, with columns of the actual matrix without missing values $S[\{c_{-\nu}\}]$. The property of $f_q = f_p$ and the score for this metric is the same as that for missing-not-at-random similarity. The implementation of this metric is as follows: We compute the correlation between a column with missing values and a categorical or numeric column without missing values. The column with missing value is represented as a binary column, where each element is either missing or non-missing. If the other column is categorical, we compute Cramer's V between the column pair; if the other column is numeric, we compute the Pearson correlation coefficient. These computations form two matrices, one for the real data and one for the synthetic data. The score is the sum of the absolute element-wise difference between these two matrices, normalized to range between $[0, 1]$.

### A.6 PRIVACY.

We use 5 metrics for the privacy group. Two are for categorical target columns: **CategoricalCAP**, **CategoricalNB**; and three are for numeric target columns: **NumericalLR**, **NumericalMLP**, **NumericalSVR**. All of them are implemented by SDMetrics. As is for the LOO metrics, each metric is repeated on each admissible target column and then averaged. Although the privacy metrics are also under the leave-one-out conditional substructure, they are more computationally expensive than the ML efficacy metrics because of the additional search step to find similar records between the two real and synthetic datasets. To restrict the computational cost, we use stratified sampling to sample about $4500$ rows from each of the real and synthetic datasets. Pilot experiments show that the change in metric values begin to diminish above this dataset size.

## B PCC

In our PCC implementation, inference proceeds via Markov chain Monte Carlo (MCMC). Column reassignment combines the slice sampler described in Ge et al. (2015) with a parallel version of the standard Gibbs sampler (Neal, 2000). Row reassignment combines the slice sampler with sequentially-allocated merge-split proposals (Dahl, 2003). We can achieve better mixing by requiring columns to use component distributions with conjugate priors and closed-form posteriors, which allows the component parameters, $\theta$, to be marginalized. For example, we model continuous features with Gaussian component models with Gaussian prior on the mean and scaled inverse chi-squared prior on the variance; and we model categorical features with categorical component models with a symmetric Dirichlet prior on the category weights.

To compute or sample from conditional distributions, $p(x|y)$, where $y$ is a set of conditions on features other than those in $x$, we use

$$p(x|y) = \prod_{u=1}^{|v|} \sum_{k=1}^{|c_u|} \left[ \pi_u(\{y_j; v_j = u\}|k) \prod_{\{j; v_j = u\}} f(x_j|\theta_{k,j}) \right],$$

$$\text{where } \pi_u(y|k) \propto |\{i; c_{u,i} = k\}| \prod_{y_j \in y} p(y_j|\theta_{k,j}).$$

To simulate $x$ without conditions, we proceed as we would with any mixture model: we choose a component, $k$, with probability proportional to the number of data assigned to it, then for each feature, $j$, we draw from $\theta_{k,j}$. To simulate $x$ with conditions $y$, we draw the component index, $k$,

with probability proportional to the number of data assigned to the component multiplied by the likelihood of the conditional under that component.

Inference of missing-not-at-random values is implemented by coupling a parent column to a binary column modeled as a mixture of Bernoulli distributions. The binary data in the Bernoulli column indicates the presence of the datum in each cell of the parent column. To compute the probability that cell $(i, j)$ is missing, one computes the likelihood of `false` under component $c_{v_j,i}$ of the Bernoulli column attached to feature $j$.

## C  DATA SYNTHESIZERS

### C.1  SYNTHPOP

Synthpop (Nowok et al., 2016) can be downloaded at `https://www.synthpop.org.uk/`. We adapted the examples in the documentation to generate synthetic data. All categorical columns are preprocessed by a re-encoding into integer values. Columns with larger than 30 categories are converted to numeric columns to keep the computational time feasible. After training and data synthesizing, the values are decoded back to their original form.

### C.2  PCC

Our implementation of the PCC can be downloaded from `urlTBA`. For the `expedia` dataset, numeric columns that record integer counts are treated as categorical columns. Columns with missing values are manually marked in the codebook to learn the missingness distribution.

### C.3  DDPM

The code for TabDDPM (Kotelnikov et al., 2023) can be downloaded from `https://github.com/yandex-research/tab-ddpm`. We made slight modifications to the code to make it run on the Mac M1 chip. The training process required splitting the original data into a train set, a validation set, and a test set; we used a 3-1-1 random split. To instantiate a model, we followed the template of the `house` dataset in the repository (see `https://github.com/yandex-research/tab-ddpm/blob/main/exp/house/config.toml`). We made minimal modifications: (1) changed the variables `num_numerical_features`, `device`, and `d_in` to correspond to our datasets and GPU; and (2) changed the `normalization` from "quantile" to "minmax" to avoid an error in the training that may be Mac-M1 specific. These settings allowed the training and sampling to run to completion, and we did not optimize further. The training step was set to 10,000. For the `expedia` and `census` datasets, the MLoss was NaN throughout the training, giving rise to poor synthetic data, where the same value populated the column for all categorical columns, and values very positive or negative populated the numeric columns.

### C.4  GREAT

The code for GReaT (Borisov et al., 2022b) can be downloaded from `https://github.com/kathrinse/be_great`. We made slight modifications to the code to make it run on the Mac M1 chip. For preprocessing, we encoded categories into integers. For training, we used the `distilgpt2` backbone as suggested in the Quickstart. Batch size and epochs were set to 32 and 150, respectively. For data generation, we used a `max_length` of 500. For post-processing, we decode the integers back to the categories and replace any unknown categories with the most frequent category. Note that without the preprocessing, GReaT generated more out-of-category samples. For example, it could generate the entry `2014-for-19` for a date column.

### C.5  GAUSSIANCOPULA, TVAE, CTGAN, COPULAGAN

These four methods are implemented by Synthetic Data Vault (SDV) (Patki et al., 2016), which can be downloaded from `https://docs.sdv.dev/sdv/`. Since the three datasets used in this paper are also from SDV, no preprocessing was performed. We followed the documentation to train and generate samples.

# D EXTENDED RESULTS

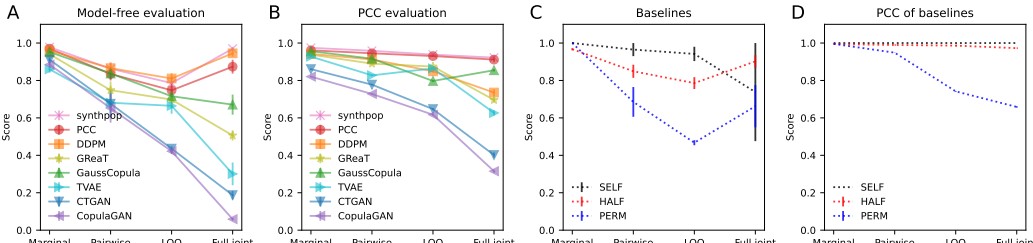

Figure S1: **Model-free and PCC-based evaluation on the student dataset. (A)** model-free evaluation of the synthesizers; **(B)** PCC-based evaluation of the synthesizers; **(C)** model-free evaluation of the baselines; and **(D)** PCC-based evaluation of the baselines. The x-axis shows the metric groups by substructure. The y-axis is the average metric score of the metrics in the group. Error bars are standard error gathered across synthetic datasets.

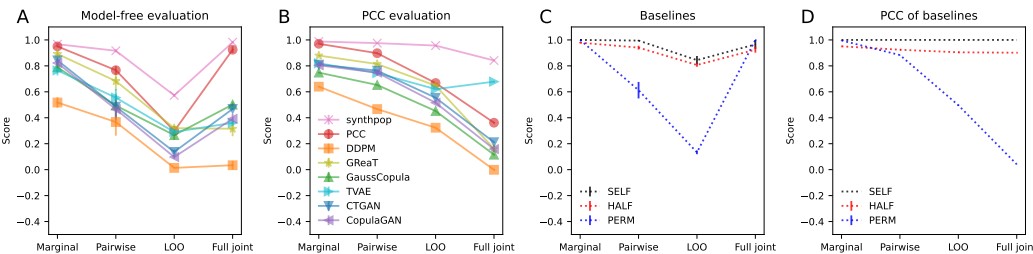

Figure S2: **Model-free and PCC-based evaluation on the expedia dataset.** See caption of Figure S1.

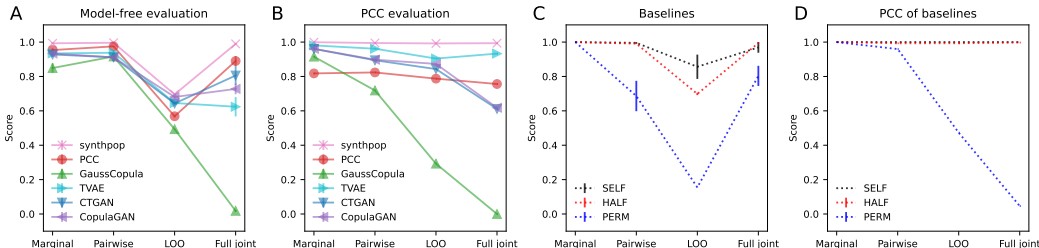

Figure S3: **Model-free and PCC-based evaluation on the census dataset.** See caption of Figure S1.

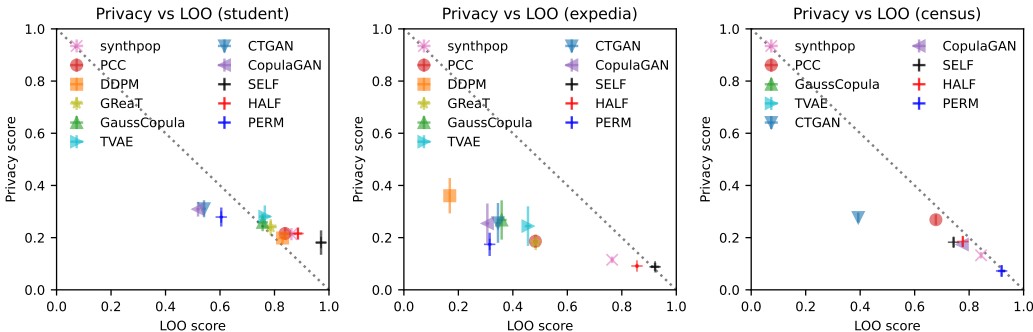

Figure S4: **Tradeoff between the LOO and privacy metric groups.** The privacy metrics target the LOO distribution but with the aim to penalize accurate inference; thus, its objective is opposite to that of the LOO metric group (Section 3). This figure confirms the almost linearly inverse relationship between the two metric groups, suggesting a strong tradeoff (`student:` $r = -0.94$; `expedia:` $r = -0.90$; `census:` $r = -0.94$). The LOO score shown is the average of the model-free and PCC-based scores, while the privacy score contains only the model-free scores.

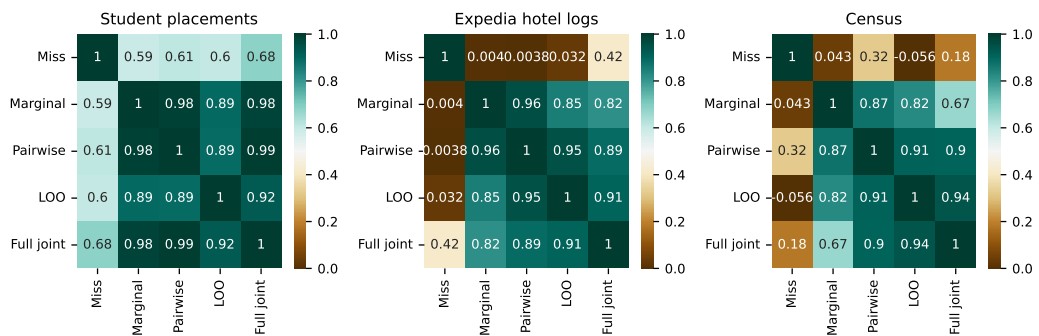

Figure S5: **Correlation between metric groups.** The correlation between missingness and the structured groups (marginal, pairwise, LOO, and full joint) is lower than the correlations among the strucutured groups themselves because the missingness metric is derived mostly from from the missingness matrix and not directly from the actual data. Furthermore, the correlation between missingenss and the structured groups tend to increase with structural complexity, potentially because missingness is more of a global metric. The correlation between the marginal and the other structured groups (pairwise, LOO, and full joint) tend to decrease, implying the existance of dependencies among the columns. Correlation values generally decrease as the dataset size increases.

## E  EXTENDED RELATED WORK

In this section we provide a detailed comparison with previous work (Dankar et al., 2022; Afonja et al., 2023) that also provide some structural organization of evaluation metrics to make clear this works novel, theoretical contributions.

Compare with the taxonomy provided by Dankar et al. (2022), we see the following similiarities: Their "attribute fidelity" and "bivariate fidelity" match the semantics of our marginal group and pairwise group, respectively. Two of their "population fidelity," namely "Distinguishability type metrics" and "The log-cluster metric," fall under our joint-distribution group. In contrast, we point out the following differences: (1) They do not define an overarching mathematical objective nor an explicit spectrum of evaluation coverage. It is only with a clearly defined objective and spectrum that one knows whether the evaluation covered the objective in full. (2) They place ML efficacy in its own category as a Narrow application-specific measure, whereas we show that ML efficacy can be interpreted as a Broad resemblance-based measure targeting a leave-one-out conditional distribution. Without this interpretation, the evaluation may require trading off between multiple objectives. With the reinterpretation, all the metrics can be ordered according to their structural complexity and be

used together in the same space, as shown in Figure 2 in the main text. (3) They place both the "cross-classification metric" and "Distinguishability type metrics" under the population fidelity, whereas we would identify the former as targeting the LOO distribution and the latter as targeting the full-joint. This distinction may be useful for metric selection and improvement. (4) They place the "Difference in Empirical distributions type metrics" under population fidelity, but our framework points out that this metric is really about the type of estimator used rather than the distributional property targeted. Again, this distinction may be useful for metric selection and improvement. (5) They place the "likelihood metrics" under population fidelity, whereas our framework has identified it as an orthogonal dimension to distributional property, and has formed an entire model-based spectrum of metrics out of it.

Compare with the taxonomy provided by Afonja et al. (2023), we see the following similiarities: Their use of the words "marginal", "pair", and "join" coincides with how we use them in our paper. Some of the "marginal" and "pair" metrics also match the metrics we use in the marginal and pairwise groups. In contrast, we point out the following differences: Same as points (1) and (2) in the previous paragraph. (3) They do not have model-based metrics. Model-free metrics avoid the issues of needing different estimators and scores for different data types and distributional properties, and are hence more coherent. (4) Their joint measure of "distance to closest record" is an interesting reinterpretation of what is typically used to evaluate privacy. Their measure of "likelihood approximation" as an extension of "distance to closest record" to the test dataset is an interesting idea for testing the joint distribution. We did not include distance-based metrics because their natural range is not [0,1]. This makes it hard to combine them with the other metrics through an average.

In general, we omitted evaluation metrics based on distances because of the range issue mentioned above, but they are surely valid metrics. To gain some insights on how distance-based metric in our framework, we highlight similarities and differences between our model-based likelihood estimate and that based on distance to closest record (DTCR) (Afonja et al., 2023). Empirically, being close to a real data point and being a likely real data point is often not the same. Consider a normal distribution. If a synthetic data point is very close to a real data point at the tail of the normal distribution, DTCR would assign it a high likelihood, whereas a gaussian model would assign it low likelihood. Conversely, if a synthetic data point is very close to the mode but there are no real data points as close to the mode, the DTCR would assign a lower likelihood than the gaussian model would. From a more theoretical perspective, the likelihood constructed from DTCR with Euclidean distance is similar to that constructed from a model built using kernel density smoothing over the training data with a Gaussian kernel of standard deviation 1. In contrast, the likelihood constructed from PCC follows Equation 3.3 in the main text. For continuous variables, PCC learns a Dirichlet Process Gaussian Mixture model with Gaussian prior on the mean and scaled inverse chi-squared prior on the variance, which can be interpreted as performing Bayesian Occam razor on the kernel density smoothing approach.

