# OpenReview forum: "Structured Evaluation of Synthetic Tabular Data"
_ICLR.cc/2024/Conference — Submitted to ICLR 2024_

### Official Review · Reviewer_9Y2W · 2023-10-24

**Soundness:** 2 fair
**Presentation:** 2 fair
**Contribution:** 2 fair
**Rating:** 5
**Confidence:** 4

**Summary:**

Tabular data generative models strive to master the underlying process that produces observed data, aiming to generate synthetic data that mirrors the observed distribution. Over the past decade, various methods, from statistical to deep-learning-based approaches, have emerged to understand these distributions. A key challenge, however, is the evaluation of the generated synthetic samples. Since the true data generating process remains unknown, measuring the effectiveness of these models is not straightforward. While many attempts have been made to standardize evaluation methodologies and to distill metrics into a consolidated framework, they often fall short in terms of objectivity and clarity in interpretation, as noted by the authors. Addressing this, the paper seeks to introduce a unified evaluation framework that consolidates current metrics under a single mathematical objective. This objective is built on the premise that synthetic data are drawn from the same distribution as the observed data. To further bolster their evaluation approach, the authors suggest leveraging the probabilistic cross-categorization method as a stand-in for the elusive ground truth of the data generating process.

**Strengths:**

I find the authors incorporation of Probabilistic Cross-categorization— a domain-general method designed for characterizing the full joint distribution of variables in high-dimensional datasets— particularly intriguing, especially in the realm of tabular data generation. This is my first encounter with this approach in a benchmarking context, and its novelty in the author's work is commendable.

**Weaknesses:**

I commend the authors efforts in detailing various metrics and particularly the authors exploration into the nuances between model-free metrics and the PCC-based metric. A deeper elaboration on this distinction would be immensely helpful for readers to fully grasp the nuances.

The representation in Figure 1B, specifically regarding the spectrums, may benefit from further context or an enriched explanation. This raises a query: Are the authors implying that model-free evaluations, such as those estimating marginal or pairwise relationships, may not provide a holistic perspective? Is there an inherent advantage in adopting model-based techniques, like PCC, to act as proxies for real data while assessing the same metrics? Moreover, given that PCC operates as a synthetic model, does its role in the evaluation process imply a comparison between synthetic models through another synthetic standard? Gaining clarity on these nuances would greatly enhance understanding.

It would also be illuminating to discern how this work either mirrors or deviates from established frameworks in previous literature. While the authors' initiative to broaden the metrics spectrum and introduce a surrogate model approximating real-data probability distribution is commendable, elaborating the distinct facets or innovative insights of the author's proposal, especially vis-à-vis findings in [1, 2] referenced in Questions section, could accentuate the originality and significance of the research amidst prevailing knowledge.

**Questions:**

General comments & questions
=========================

- In section 3, the authors mentioned that “the objective of the data synthesiser is \( Q=P \)”. While I understand the underlying objective might be to highlight the close similarity between the distributions, stating it in this manner might lead some readers to interpret this as \( Q \) being an exact replica of \( P \). Given the paper's central theme of using \( Q \) as a more private alternative to \( P \), such an interpretation could be seen as contradictory. Perhaps it might be clearer to emphasize that \( Q \) is intended to be statistically analogous or mirrors the distribution of \( P \). This would signify that while \( Q \) captures the broader statistical characteristics of \( P \), individual data points might differ, ensuring privacy.  I believe a more detailed description or clarification in this section could be beneficial for enhancing the reader's understanding and mitigating potential misconceptions.

- The presentation of the leave-one-out (LOO) metric seems to bear a resemblance to the dimension-wise prediction performance metric as described in references [3, 4], as well as the all model's test metric outlined in [2]. Could the authors clarify whether these are synonymous or if there's a discernible distinction between them?

- Rather than depending on a surrogate model to estimate ground truth, would it not be more reliable to employ a distinct hold-out test set, ensuring it retains the same distribution as the real (observed) data? Admittedly, this approach might pose challenges when dealing with limited samples. However, in such scenarios, methodologies like k-fold validation could be explored to compute an average score over several iterations. Alternatively, having a baseline that shows the performance of the surrogate on hold-out test set could serve as the acceptable error threshold.

- The current presentation of details incorporates a variety of symbols, which, while comprehensive, can sometimes add complexity to the narrative without necessarily enhancing clarity. To improve readability and facilitate a deeper understanding for readers, I'd recommend introducing a dedicated subsection early on to familiarize readers with the notation. This way, within the main text, the authors can focus on using notation only when it brings forth novel information, and rely on plain language descriptions when the content permits. For instance, the passage: "We then use the surrogate model to compute   \{ \hat{P(X_i) \mid i=1,..,n \}, which is the likelihood of  X_i…" could be more intuitively conveyed as: "We use the surrogate model to determine the likelihood of the real data samples under this model." If the precise mathematical formulation is essential to the discussion, consider placing it in a distinct equation block, which can then be easily referenced within the narrative.

- In section 3.3, the discussion surrounding the pp-plot could benefit from further clarity. I was wondering if the likelihood estimate method introduced is akin to the "Distance to Closest Record" concept mentioned in [5], where a Nearest Neighbours model is employed to gauge the likelihood of each synthetic data originating from the real data distribution. Is the primary distinction here the use of the Probabilistic Cross-Categorisation model? Any elucidation on this comparison would be invaluable for readers familiar with the referenced methodology.


- Given that the evaluation encompasses a diverse range of metrics within the same family, such as marginal, pairwise-based, and leave-one-out conditionals, full-joint-based, missingness, and privacy. It might be insightful for readers if a correlation plot is provided. Such a plot could help elucidate potential correlations among metrics both within the same group and across different groups. This added visual representation could offer a comprehensive perspective on the interplay of these metrics and their potential overlaps or distinctions.


Small typo
====

Figure 1 (A) Model fee -> Model free

(Potential) missing reference
======================

It appears there's an omission in the paper's review of related literature. In particular, ref. [2] in its section 3 emphasizes the significance of evaluating synthetic tabular data generators across various metrics, including marginal-based, column-pairs, joint, and utility considerations. The thrust of these discussions in [2] bears a strong resonance with the core objectives of this paper. It's surprising and noteworthy that such pertinent work isn't cited or discussed in the current paper's related work section.

References
=========

[1] Dankar, F.K., Ibrahim, M.K. and Ismail, L., 2022. A multi-dimensional evaluation of synthetic data generators. IEEE Access, 10, pp.11147-11158.

[2] Afonja, T., Chen, D. and Fritz, M., 2023. MargCTGAN: A" Marginally''Better CTGAN for the Low Sample Regime. arXiv preprint arXiv:2307.07997.

[3] Choi, E., Biswal, S., Malin, B., Duke, J., Stewart, W.F. and Sun, J., 2017, November. Generating multi-label discrete patient records using generative adversarial networks. In Machine learning for healthcare conference (pp. 286-305). PMLR.

[4] Engelmann, J. and Lessmann, S., 2021. Conditional Wasserstein GAN-based oversampling of tabular data for imbalanced learning. Expert Systems with Applications, 174, p.114582.

[5] Zhao, Z., Kunar, A., Birke, R. and Chen, L.Y., 2021, November. Ctab-gan: Effective table data synthesizing. In Asian Conference on Machine Learning (pp. 97-112). PMLR.

---

> ### Author Response · Authors · 2023-11-17
>
> We thank the reviewer for the many detailed comments that help improve the paper. In particular, we thank the reviewer for recognizing the novelty of the PCC-based metric and encouraging us to elaborate on it.
>
> **Elaboration of the PCC-based metric:**
> We agree that it is helpful to have a deeper elaboration on the distinction between model-free and the PCC-based metric, as well as an enriched explanation on why one would like to have the model-based metrics. To achieve that, we will add a summary of our responses below to the Introduction and Section 3. The response below is also our point by point response to the specific questions raised by the reviewers.
>
> Whether the evaluation is holistic or not depends on whether the metrics provide coverage of the *full joint distribution*. This means that evaluating only marginal and pairwise relationships would not provide a holistic perspective, because higher order statistics such as 3-way interactions would not be captured by the evaluation. Using metrics that target the full joint distributions (i.e., LOO or full-joint) could in principle provide a holistic picture. This is true regardless of whether the metrics are model-free, model-based, or a combination of the two. The issue with the model-free metrics is that they are often noisy, especially towards the RHS of the spectrum for the LOO and full-joint metrics, as shown by the large error bars and non-monotonicity in panels A and C of Figures S1–S3. The large error bars and non-monotonicity originate from (1) the use of multiple ML models with different expressiveness, as well as (2) the different types of estimators and scores used for different distributional properties and data types. For example, the full-joint metrics were evaluated indirectly via a discriminator, which also included a linear version (i.e., logistic regression) and a simple nonlinear one (i.e., SVC). In other words, the model-free metrics are *incoherent* in terms of the estimators (e.g., the underlying ML models used) and scores. In contrast, the model-based metrics neither have the incoherence problem, nor exhibit the noisy and non-monotonicity, as evidenced in panels B and D of Figures S1–S3. Thus, the inherent advantage of the model-based technique is the coherence in methodology, that is, all the metrics can be derived from the same type of estimator and score.
>
> The reviewer is correct that the model-based technique puts the evaluation on another synthetic standard. The worry is then whether this synthetic standard would bias the evaluation. The degree of bias depends on how much the surrogate model would mask statistical properties because of a lack of expressiveness. To illustrate, if the surrogate model captures only the marginal distributions but not the pairwise and higher-order dependencies, it would assign similar likelihoods to the PERM baseline and the real dataset. We have two pieces of evidence that PCC does not suffer from such lack of expressiveness. First, PCC is consistently a top data synthesizer as evaluated by the model-free metrics (panel A in Figure S1–S3). Second, the model-based evaluation derived from PCC does not always put itself as the highest performing model but can capture superiorities of other techniques (panel B in Figure S1–S3). In fact, the ordering of the performance of the different data synthesizers is consistent between the PCC-based and model-free evaluations.

---

> > ### Author Response · Authors · 2023-11-17
> >
> > **Compare and contrast with refs [1,2]**:
> > We thank the reviewer for bringing ref [2] to our attention. (Ref [1] was cited already.) Below we make a vis-à-vis comparison between our approach and refs [1,2].
> >
> > What we and ref [1] have in common:
> > - The “attribute fidelity” and “bivariate fidelity” in ref [1] match the semantics of our marginal group and pairwise group, respectively.
> > - Two of the “population fidelity” in ref [1], namely “Distinguishability type metrics” and “The log-cluster metric,” can fall under our joint-distribution group.
> > What we have that ref [1] does not have:
> > 1. Ref [1] does not define an overarching mathematical objective nor an explicit spectrum of evaluation coverage. It is only with a clearly defined objective and spectrum that one knows whether the evaluation covered the objective in full.
> > 2. Ref [1] places ML efficacy in its own category as a Narrow application-specific measure, whereas we show that ML efficacy can be interpreted as a Broad resemblance-based measure targeting a leave-one-out conditional distribution. Without this interpretation, the evaluation may require trading off between multiple objectives. With the reinterpretation, all the metrics can be ordered according to their structural complexity and be used together in the same space, as shown in our Figure 2.
> > 3. Ref [1] places both the “cross-classification metric” and ““Distinguishability type metrics” under the population fidelity, whereas we would identify the former as targeting the LOO distribution and the latter as targeting the full-joint. This distinction may be useful for metric selection and improvement.
> > 4. Ref [1] places the “Difference in Empirical distributions type metrics” under population fidelity, but our framework points out that this metric is really about the type of estimator used rather than the distributional property targeted. Again, this distinction may be useful for metric selection and improvement.
> > 5. Ref [1] places the “likelihood metrics” under population fidelity, whereas our framework has identified it as an orthogonal dimension to distributional property, and has formed an entire model-based spectrum of metrics out of it.
> >
> > What we and ref [2] have in common:
> > - The use of the words “marginal,” “pair,” and “joint” in ref [2] coincides with how we use them in our paper. Some of the “marginal” and “pair” metrics also match the metrics we use in the marginal and pairwise groups.
> > What we have that ref [2] does not have:
> > - Same as points 1 and 2 above.
> > - Ref [2] does not have model-based metrics. Model-free metrics avoid the issues of needing different estimators and scores for different data types and distributional properties, and are hence more coherent.
> > What ref [2] has that we do not have:
> > - Their joint measure of “distance to closest record” is an interesting reinterpretation of what is typically used to evaluate privacy. Their measure of “likelihood approximation” as an extension of “distance to closest record” to the test dataset is an interesting idea for testing the joint distribution. We did not include distance-based metrics because their natural range is not [0,1]. This makes it hard to combine them with the other metrics through an average.
> >
> > In summary, our work is unique in that we provide a clear objective and spectrum of evaluation which, for the first time, allows us to:
> > - reason about the completeness of the metrics;
> > - reason about the coherence of the all the metrics, not as a multidimensional evaluation, but as a structured evaluation where we can order, visualize, and combine all the metrics in a principled way;
> > - develop a spectrum of coherent model-based metrics that are free from the need to use different estimators and scores for different data types and distributional properties.
> >
> > We will summarize these points related to refs [1,2] in the Related Work section with a pointer to a new Appendix E, work the above summary points into the Introduction, and update Figure 1 to show that the “Population metrics” in ref [1] spreads across LOO and the full-joint group.

---

> > > ### Author Response · Authors · 2023-11-17
> > >
> > > **Clarifying Q = P but S $\neq$ X**:
> > > Following the reviewer’s suggestion, we have changed the first sentence in the first paragraph of Section 3 to say: “The goal of a data synthesizer is to produce samples that are statistically analogous to the real data but not direct copies of them.” We have also revised the paragraph to put “Q = P” and “S $\neq$ X” right beside each other. Note that in the original paper we already tried to make this point. For example, in the Introduction we say “Our framework stems from a formal objective that a synthesizer should produce *samples from the same joint distribution* as the real data.” In the privacy subsection, we say “While we would like to have Q = P , we do not want the synthesizer to simply memorize the training data, resulting in S = X.”
> > >
> > > **On refs [3,4]**:
> > > We thank the reviewer for pointing out the column-wise prediction in refs [3,4]. Column-wise prediction as presented in [3] is surely a kind of LOO. The difference between dimension-wise prediction and ML efficacy is the following: Let the model trained on the synthetic data be $M_S$, and the model trained on the real data be $M_R$. In column-wise prediction the comparison is between the predictions of $M_S$ and $M_R$, whereas in ML efficacy the comparison is between the predictions of $M_S$ and the corresponding column of the real data. Also, ML efficacy does not require splitting the data into a train and a test set, while the dimension-wise prediction uses a split. We will be happy to include column-wise prediction into the paper if this is crucial for increasing the rating of the paper.
> > >
> > > **On hold-out test set**:
> > > We are not sure how a hold-out test set would function in place of the surrogate model. The main difficulty in evaluating synthetic data lies in the difficulty to compare two datasets directly, but the PCC as a surrogate model can summarize any subset of a dataset with a number (the subset’s likelihood) to allow direct comparison between datasets. If the reviewer is referring to the dimension-wise prediction of [3,4], we agree that having a hold-out test set is a valid approach for evaluating the LOO distribution. Or perhaps the reviewer is referring to the “likelihood approximation” in [2], which is surely a valid model-free approach for evaluating the joint distribution. In fact, the “likelihood approximation” may be a more bonafide model-free approach, as it relies only on computations of Euclidean distances instead of training a discriminative model. The performance of the surrogate on a hold-out set can be seen in Figures S1–S3’s panel D with the HALF baselines. Please let us know if we are misunderstanding the question.
> > >
> > > **On improving readability and clarity**:
> > > We thank the reviewer very much for the suggestion to improve the readability paper with such a detailed example! Following the suggestion, we will separate as much as possible the plain English description from the mathematical notation, and lead with the plain English description.
> > >
> > > **On likelihood estimate via Distance to Closest Record**:
> > > We thank the reviewer for asking this interesting question. There are several major distinctions between our model-based likelihood estimate and that based on distance to closest record (DTCR). Empirically, being close to a real data point and being a likely real data point is often not the same. Consider a normal distribution. If a synthetic data point *s* is very close to a real data point at the tail of the normal distribution, DTCR would assign *s* a high likelihood, whereas a gaussian model would assign it low likelihood. Conversely, if a synthetic data point is very close to the mode but there are no real data points as close to the mode, the DTCR would assign a lower likelihood than the gaussian model would. From a more theoretical perspective, the likelihood constructed from DTCR with Euclidean distance is similar to that constructed from a model built using kernel density smoothing over the training data with a Gaussian kernel of standard deviation 1. In contrast, the likelihood constructed from PCC follows Equation (1) in the paper. For continuous variables, PCC learns a Dirichlet Process Gaussian Mixture model with Gaussian prior on the mean and scaled inverse chi-squared prior on the variance (see Appendix B), which can be interpreted as performing Bayesian Occam razor on the kernel density smoothing approach. We now mention this in the Related Work with a pointer to the more detailed discussion in Appendix E.
> > >
> > > **Others**:
> > > Following the reviewer’s suggestion, we will show the correlation plots of the individual metrics within each structural group as well as correlation plots of the group metrics across the structural groups. These will be added to Appendix D and referred to in Section 5.
> > >
> > > We thank the reviewer for catching the typo in Figure 1A. It has been fixed.
> > >
> > > We now include references [2,3,4,5] in the paper. Reference [1] was in the paper already.

---

> > > > ### Comment · Reviewer_9Y2W · 2023-11-20
> > > > **Response**
> > > >
> > > > I thank the authors for addressing my comments and for their contributions. I have reviewed the changes and my evaluation remains unchanged.

---

> > > > > ### Author Response · Authors · 2023-11-20
> > > > >
> > > > > We just updated the PDF with all the changes we mentioned in our response. Sorry for the delay between the response and the PDF update.
> > > > >
> > > > > The updated texts are in blue for ease of tracking. We added the correlation plot between metric groups to Figure S5 in Appendix D. The correlation plot within metric group is visually messy, so we decided to leave it out. A newly added Appendix E includes the comparison between our work and refs [1,2].
> > > > >
> > > > > We are sorry for the confusion caused by the delay in PDF update, and hope that the implemented changes are adequate.

---

> > > > > > ### Comment · Reviewer_9Y2W · 2023-11-22
> > > > > > **Response**
> > > > > >
> > > > > > I appreciate the authors' efforts in updating their paper with the suggested changes, which have notably improved its quality. After reviewing the revisions, I find the paper to still be quite dense. A further round of editing could elevate it even more. While I am maintaining my initial score, I am optimistic about the paper's future iterations and commend the authors for their dedication to enhancing their work.

---

### Official Review · Reviewer_UGhN · 2023-10-29

**Soundness:** 3 good
**Presentation:** 3 good
**Contribution:** 3 good
**Rating:** 6
**Confidence:** 4

**Summary:**

In this paper, the authors propose an analysis framework for evaluating data synthesizers. Data synthesizers aim to create synthetic datasets that resemble real datasets without directly copying them, i.e., the goal of such synthesizers is to generate synthetic datasets of a distribution Q that is as close as possible to the distribution P of the real dataset. The authors have conducted a structured evaluation of SOTA techniques for data synthesis on different datasets for varying evaluation criteria for distributions ranging from Missingness to Univariate Marginal to Pairwise Joint to Leave-One-Out conditionals to Full Joint Distribution.

**Strengths:**

The topic of data synthesis is highly relevant for many real-world applications where data is very costly to obtain or privacy is a major concern.
The presentation is well-structured and detailed.
The authors have taken a systematic approach to evaluate different synthesizers in comparative way. They have considered different metrics and provided clear explanations for their choices.

**Weaknesses:**

Although well-structured, the presentation is quite dense, and it might be challenging for someone without a background in the area to understand the differences and significance of the analysis framework and findings.
The paper has a strong focus on the technical evaluation of synthesizers, but it doesn't discuss the practical implications of the findings. I.e., how might these results impact real-world applications of these synthesizers?
It would be useful to know how the methods would have performed on large-scale datasets if computational resources were not a constraint.
While the proposed metrics focus on a quantitative evaluation, qualitative insights or user-based evaluations might provide a more holistic view of synthesizer effectiveness.

**Questions:**

See weaknesses above.

**Details Of Ethics Concerns:**

None.

---

> ### Author Response · Authors · 2023-11-17
>
> We thank the reviewer for noting that our approach is systematic and our evaluation is comparative, as well as for all the comments that help improve the paper.
>
> To improve the readability of the paper, we now make a cleaner separation between plain English and math notations throughout Section 3 and Appendix A. We hope that this would help lessen the cognitive load and make the reading feel less dense.
>
> We thank the reviewer for asking the question about practical implications and impact on real-world applications. We provide the following main points:
> - Synthesizer selection: The structured evaluation provides a complete and coherent picture of the performance of different synthesizers on a given dataset. This allows a practitioner to select the best data synthesizer with confidence, knowing the degree and extent to which the full joint is mimicked. The HALF baseline also provides a rough estimate on how far the data synthesizer is from being very good.
> - Synthesizer improvement: The structured evaluation with the baselines show where along the structural spectrum a data synthesizer falls short. This could expose particular failure modes and facilitate better design of the data synthesizer.
> - Evaluation improvement: The structured spectrum suggests that one could design many new metrics, such as 3-way interactions and leave-n-out (as suggested by Reviewer fbzR), and know where to position them relative to the other metrics. Furthermore, one could investigate the error bars and non-monotonicity of the ordered metric groups to identify problematic metrics. Finally, thinking about the chain of sufficiency could help the selection of estimators to use (e.g., which ML models to use in ML efficacy).
> - Human-centered debugging: The structured framework provides a nice, ordered visualization for people to spot strange behaviors more easily. An example is given at the end of this response related to the comments on “qualitative insights or user-based evaluations.”
> We will add a summary of the above points towards the end of the paper.
>
> We echo the reviewer’s sentiment on testing our approach on large-scale datasets. This is of course difficult to realize because computational resources are real constraints. We would also like to point out that the census dataset we analyzed is considered a sizable tabular dataset. Many seemingly larger datasets are larger because the categorical columns have been preprocessed to use the one-hot encoding. Nevertheless, by looking at the three datasets of different sizes analyzed in the paper (Figure 2), we can already make some qualitative inferences about the performance of the methods:
> - Deep-learning methods generally become better as the dataset size increases.
> - Methods that are not designed to capture the full-joint distribution (i.e., GaussianCopula) generally become worse as the dataset size increases.
> - The performance of structurally based methods (synthpop and PCC) seems to be less dependent on dataset size.
> Some of these were mentioned in the paper on page 8, but we will make sure all the above points are covered.
>
> We definitely agree with the reviewer that “qualitative insights or user-based evaluations might provide a more holistic view of synthesizer effectiveness.” We think that the structured evaluation would aid user-based evaluations by helping the users to more quickly drill down to where the problems might be. For example, in the updated Figure 2 first row, second column, GReaT has a suspiciously low performance in the Missingness group. One can then drill down to see that the cause is due to GReaT generating missing values in columns that do not have any missing values in the original data (mentioned in page 8, second last paragraph of the original manuscript).

---

> > ### Comment · Reviewer_UGhN · 2023-11-22
> > **Thank you for the response.**
> >
> > I thank the authors for their comprehensive response. Although I am very much in favor of research focusing on evaluation strategies and frameworks, it is crucial that such research provide clear practical and theoretical insights/contributions. For now, the insights gained from your evaluation framework seem rather vague. Therefore, I will keep my initial score.

---

### Official Review · Reviewer_fbzR · 2023-10-29

**Soundness:** 2 fair
**Presentation:** 1 poor
**Contribution:** 3 good
**Rating:** 3
**Confidence:** 4

**Summary:**

The paper proposes a novel evaluation framework for evaluating tabular data generators. The framework has been generated with a single mathematical objective, i.e., that a data generator should produce samples from the same joint distribution as the real data. The framework includes both existing and novel metrics which are then arranged along a  spectrum which holds for both model-free and model-based metrics.

**Strengths:**

The main strength of the paper is that such work is very much needed. Additionally, I like the idea of arranging the metrics according to a spectrum that highlights the complexity of the relationships among features that a metric is able to capture.

**Weaknesses:**

I think the paper needs a lot of rewriting (and probably more space, I would suggest to the authors to submit to a journal).
At times it is quite difficult to follow and a lot of the metrics that are presented in Table 1 are not covered at all in the main text.
Also, it is not feasible to think that one will evaluate their models according to all the metrics shown in Table 1. A significant contribution would be to identify different subsets of these metrics and show how to use them together to capture all the desired properties of the system (see, for example, what was done for multi-label classification problems in [1]).

Also, I have some questions regarding how the ML efficacy belongs to the substructure "leave-one-out conditional distribution". Indeed, in order to *leave-one-out* then you assume that the target is a single column. In many cases, the target might not be a single column. How would this affect your thesis? Even more importantly, you write that the implication holds if we do this *for all* $j$s (i.e., by leaving out all columns). The ML efficacy test leaves out a single column, so how do you get sufficiency in this case? Finally, to define the score you use the function argmax, what happens if the problem is a regression problem or binary?

Regarding the full joint substructure you write: "Sufficiency is hard to show because...". Is it sufficient? Is it necessary?
In general, for all these substructures, I would have liked to see a much more structured approach to showing sufficiency and necessity.

I am also not sure whether I fully understand the properties of the HALF baseline. Could you please clarify why is it useful and why it provides an upper bound? Also, do you have some proof of the upper and lower bounds provided by the baselines?

At a certain point on page 6, the authors write: "We then use pp-plot to compare these two lists of probabilities." What is pp-plot? Why is it used?

In Table 2, why is Quality not reported for 2 models on the census dataset?


Minor comments:
- there is a typo on page 5 in the Missingness paragraph: $Q_v(c_j) = P_v(c)j)$
- add a bird-eye view of subsection 3.2 rather than just starting to describe baselines one by one
- put tables in the right format (i.e., as large as text)


[1] Spolaor, N., Cherman, E. A., Metz, J., & Monard, M. C. (2013). A systematic review on experimental multi-label learning. Tech. rep. 362, Institute of Mathematics and Computational Sciences, University of Sao Paulo.

**Questions:**

See above

---

> ### Author Response · Authors · 2023-11-17
>
> We thank the reviewers for the helpful comments to make the paper more accessible and useful.
>
> **On rewriting and implementation feasibility**:
> To help with readability, we will revise Section 3 such that the intuitive description in plain English is well separated from complex mathematical notation. We now also make sure the main text mentions all the metrics in Table 1 with references to the Appendix. Our intention in presenting Table 1 was not to provide the recipe for constructing a structured evaluation but to provide a summary of the structured framework and what it covers. Upon acceptance, we will open source our implementation, as mentioned in the list of contributions, so that any reader can use and build on it, and to ensure feasibility. To emphasize this point, we add in the caption to Table 1 that the code is at <url TBA>. Given this readily available codebase, we believe it reasonable and important to use ALL the metric groups instead of a subset of them. Section 5 and Figure 2 show precisely how all the metrics can work together to form a coherent and complete picture of the evaluation via the naturally ordering induced by the structured framework.
>
> **On ML efficacy and LOO**:
> We thank the reviewer for bringing up these questions. If the target includes multiple columns, we can simply generalize the leave-one-out (LOO) to leave-n-out (LNO). The implication on how LNO is related to the objective of Q = P is the same as LOO, because all the dependencies described by the chain rule factorization would still be covered. Its natural ordering would be after the LOO and before the full joint when added to Figure 2. The reviewer is correct that if ML efficacy is tested on a single column, it would not be sufficient. This is why in our implementation we loop through all the columns. We now make a note that this is a slight generalization from how people usually use ML efficacy in the Leave-one-out subsection. Lastly, the binary problem is a special case of the classification problem with 2 classes, so the argmax would cover the binary case. Appendix A.3 lists the ML models used for both the binary and categorical classification. The reviewer is correct that the argmax would not be used for a regression problem. We did not include ML regressors because their measure of accuracy (i.e., RMSE) does not fall inside the range of [0,1], and hence, not in the right scale to be averaged with the other metrics. To fill this lack, the model-based LOO metrics is agnostic to column types and do cover all the columns. This is a strength of the model-based metrics that we now highlight explicitly in various parts of the paper.
>
> **On sufficiency**:
> We thank the reviewer for pointing out the vagueness about the sufficiency related to the joint-distribution metric. Because we cannot show sufficiency, we now say it is likely not sufficient because there may be multiple ways that the implicit $f_q$ and $f_p$ can produce a discriminator to achieve perfect ROC AUC. The condition is necessary for Q = P, which we mention in the main text just before the text quoted. The approach we show necessity and/or sufficiency was outlined in the “Sketch of analysis” subsection in the beginning of Section 3. We now add a few sentences in that sketch to describe in more detail our approach to show necessity and/or sufficiency.
>
> **On the HALF baseline**:
> The idea of the HALF baseline is to split the real dataset into two, and then run the evaluation to compare the two datasets. It is useful because this ensures that both datasets are from the real data generating distribution, yet not direct copies of each other. In other words, the HALF baseline is the closest thing to meeting the proposed objective of Q =P and S $\neq$ X. Thus, it should provide the desirable target to the metrics. We agree that “upper bound” is probably not the right word for the HALF baseline, because the SELF baseline beats it. We now change the “upper bound” associated with the HALF baseline to the “target value” in the text. For the SELF baseline, the upper bound on fidelity metrics follows from the fact that the synthetic and real distributions are equal (Q=P), and the lower bound on the privacy metrics follows from the fact that the synthetic dataset is a direct copy of the real dataset (S=X). How the metrics relate to Q=P is summarized in the Implementation column in Table 1. Also, the second row of Figure 2 shows that empirically, the SELF baseline is an upper bound, and the HALF baseline is also higher or equal to the methods.
>
> **On pp-plot**:
> A pp-plot is a plot that plots two cumulative distributions against each other. It is used to detect how different the two distributions are. If the two distributions are the same, the plot will follow the x=y line. We will revise Section 3.3 to clarify this.

---

> > ### Author Response · Authors · 2023-11-17
> >
> > **Others**:
> > We omit GReaT on the census dataset because we were not able to finish training given our computational resources. We omit DDPM on the census because of its poor quality (always generating the same category for categorical columns and extreme values for continuous columns). We mention this in the caption of Table 2, but now also mention it in the caption of Figure 2.
> >
> > We have fixed the typo on page 5. Thank you.
> >
> > We have added an introductory sentence to Section 3.2 as suggested.
> >
> > We have restructured Table 1 to fit the text width by removing the Target substructure column and placing it as rows. We have also combined some metrics into one and simplified the estimates to describe f_q and not f_p for brevity. To make Table 2 fit the text width, we removed the decimals if the figure is greater than 3 digits.

---

### Author Response · Authors · 2023-11-17
**Global response**

We thank the reviewers very much for their detailed, insightful, and helpful comments.

All the reviewers mentioned issues with the readability of the paper stemming from the density of information. To address these issues, we will (and have done some) rewrite much of Section 3, focusing on separating plain English description from the mathematical notation to consolidate and minimize the latter. We have also simplified Table 1 and made sure the metrics are mentioned in the main text, as suggested by Reviewer fbzR.

We thank Reviewers UGhN and 9Y2W for the inspiring comments, which help us sharpen the contribution of the paper. We will incorporate sharpened statements related to (1) the uniqueness of our structured framework, (2) the inherent advantage of our proposed PCC-based metrics, and (3) the practical implication and impact of our work.

Lastly, we would like to point out that the main novelty and contribution of the paper is the completeness and coherence of the framework, and less about the specific metrics we included. Our structured framework allows one to reason about what even constitutes a complete set of evaluation metrics. Without such a framework, there is no direct answer to whether the metrics one is using is enough, even in principle. Our structured framework also allows one to reason about the coherence of all the metrics, which determines how the metrics should be used together and helps expose shortcomings in the synthesizers or the metrics themselves. Without the framework, the field often considers each metric as its own dimension and sometimes mixes up metrics along different dimensions as the same type of metric. Lastly, the structured framework gives us the clarity to develop a spectrum of model-based metrics that are coherent in the estimators and scores used, unlike the model-free metrics that use different estimators and scores for different data type and distributional properties.

Below in the local responses we respond to all the reviewers’ comments and questions in detail.

---

> ### Author Response · Authors · 2023-11-20
> **PDF updated**
>
> We have just updated the PDF. The updated version now includes all the changes we mentioned in the responses. Updated text are colored in blue (except for the content in Table 1).

---

### Meta-Review · Area_Chair_RMAg · 2023-12-18

**Metareview:**

The paper proposes a framework to *evaluate* synthetic tabular data. Several key concerns by the reviewers include the denseness of the paper, which needs to be refined further (9Y2W) to become clearly understandable, maybe submit to another venue / journal (fbzR).

I am concerned whether the LOO => LNO argument by the authors following a key comment by fbzR would be really feasible for realistic tabular data. The authors claim in the first sentence of the abstract that tabular data is "small in volume", which perhaps is true for the UCI domains but definitely plainly inaccurate for many real-world (industrial) domains. I wonder whether the argument for LNO would be easily deployable.

Overall, the paper's objective is commendable but its approach definitely needs to be updated following the comments by the reviewers.

**Justification For Why Not Higher Score:**

Weak paper (see reviews).

**Justification For Why Not Lower Score:**

N/A

---

### Decision · Program_Chairs · 2024-01-16

Reject